# Effects of Lactic Acid Bacteria-Inoculated Corn Silage on Bacterial Communities and Metabolites of Digestive Tract of Sheep

Hongyan Han [1,†], Chao Wang [2,†], Zhipeng Huang [3], Ying Zhang [1], Lin Sun [2], Yanlin Xue [2,*] and Xusheng Guo [3,*]

1. State Key Laboratory of Reproductive Regulation and Breeding of Grassland Livestock, School of Life Sciences, Inner Mongolia University, Hohhot 010070, China; hanhongyan1018@163.com (H.H.); 13674745953@163.com (Y.Z.)
2. Inner Mongolia Engineering Research Center of Development and Utilization of Microbial Resources in Silage, Inner Mongolia Academy of Agriculture and Animal Husbandry Science, Inner Mongolia Key Laboratory of Microbial Ecology of Silage, Hohhot 010031, China; wangchao200612@hotmail.co.jp (C.W.); linsun@cau.edu.cn (L.S.)
3. State Key Laboratory of Grassland and Agro-Ecosystems, School of Life Sciences, Lanzhou University, Lanzhou 730000, China; huangzhp2017@lzu.edu.cn
* Correspondence: xueyanlin0925@outlook.com (Y.X.); guoxsh07@lzu.edu.cn (X.G.)
† These authors contributed equally to this work.

**Abstract:** Silage is widely used as ruminant feed all over the world. Lactic acid bacteria inoculants are commonly applied in silage production to improve preservation efficiency. To investigate the effects of lactic acid bacteria-inoculated silage on the bacterial communities and metabolites of the digestive tract, twenty-four local hybrid rams (a hybrid of Small Tail Han sheep and Mongolian sheep with an average initial BW 27.8 ± 3.05 kg) were randomly divided into four groups fed with corn stalk (S), corn stalk silage, corn stalk silage treated with *Lentilactobacillus plantarum* (P), or corn stalk silage treated with *L. buchneri* (B). The results showed that compared with the control and B groups, the inoculant with P significantly increased silage dry matter (DM) content, while the pH value was significantly higher than that in group B, and the aerobic stability was significantly lower than that in group B. Firmicutes and Bacteroidetes were the two dominant phyla of digestive tract microbiota in sheep. Compared with corn stalk, sheep fed with corn stalk silage showed a higher relative abundance of *Prevotella*. P-treated silage decreased the relative abundance of Firmicutes at the phylum level in rumen fluid. Silage treated with P or B increased the relative abundance of *Prevotella*, *Ruminococcus*, and *Fibrobacter* at the genus level in the rumen. A total of 498 differential metabolites in the rumen were detected when comparing the corn stalk and corn stalk silage groups. A total of 257 and 141 differential metabolites were detected when comparing the untreated silage and silages treated with P and B, respectively. These metabolites in the sheep rumen were correlated with bacterial communities, especially *Butyrivibrio*, *Fibrobacter*, and *Prevotella*. In conclusion, the addition of P and B during ensiling could change the fermentation and metabolites in the rumen by influencing the bacterial community. The change induced by these inoculants might be beneficial for animals' performance and the health of ruminants.

**Keywords:** lactic acid bacteria; silage; rumen; bacterial community; metabolites

## 1. Introduction

It is well-known that the composition and the diversity of microorganisms in the gastrointestinal tract, especially the rumen, are of primary importance in the health and nutrition of ruminants, and ultimately influence animal performance. It is estimated that over 5000 species of microorganisms, of which 95% are bacteria, methanogenic archaea, protozoa, fungi, and bacteriophages, inhabit the rumen [1]. The main function of these

microorganisms is the decomposition of plant fibers, mainly cellulose and hemicelluloses in the diet, and producing volatile fatty acids, high-quality proteins, and vitamins to meet the maintenance and growth requirements of ruminants [2]. The gastrointestinal tract microbiota also plays an important role in immune system stimulation and pathogen bacteria resistance [3,4]. The rumen microbiota can be influenced by many factors, such as diet, breed, age, and environmental conditions. Among these factors, diet is a major and important factor in shaping the gut microbiota [5]. More and more studies have been conducted to investigate the effects of diet on the microbial diversity in the rumen. Yang et al. [6] characterized the in situ rumen degradability and microbiota of four barley grain varieties and corn and found that *Prevotella* were the predominant primary colonizers of grains. Starch digestion was accelerated upon the establishment of the secondary colonizers, with *Lentilactobacillus* predominating in mature biofilms.

Silage, the main ruminant feed fermented under anaerobic conditions, is widely used all over the world because it minimizes the total loss of nutrients from harvest through storage and allows for easier feeding, and it often allows greater efficiency and timeliness of feed mixing and handling on farms than dry forages [7]. In order to improve the fermentation quality of silage and the aerobic stability during the feeding process, microbial additives based on lactic acid bacteria are widely recognized and used [6]. Numerous studies have shown that the addition of homofermentative lactic acid bacteria can accelerate the production of lactic acid and the decline of pH and thus improve the fermentation quality and reduce the loss of dry matter. *Lentilactobacillus plantarum* has been reported to be the most commonly used silage inoculant [8]. The addition of heterofermentative inoculants can increase the content of acetic acid, inhibit the activity of yeasts and molds, and thus improve the aerobic stability of the silage. Studies have reported that feeding silage supplemented with lactic acid bacteria has positive effects on ruminants' performance. Zhang et al. [9] reported that bulls fed lactic acid bacteria-inoculated silage had a higher daily intake than the control. Cherdthong et al. [10] found that rice straw silage treated with *Lentilactobacillus casei* showed higher digestibility for beef cattle. Basso et al. [11] found that silage inoculated with *L. plantarum* and *L. buchneri* improved the average daily gain of lambs. Monteiro et al. [12] also reported that alfalfa silage treated with *L. plantarum* improved milk production in high-producing dairy cows. However, the effects of lactic acid bacteria-inoculated silage on the gastrointestinal tract microbiota are still unclear.

According to reports of "OECD-FAO Agricultural Outlook 2020–2029", with the increase in population and the demand for milk, dairy products, and beef, the rearing of ruminants is becoming more and more important in agricultural production. Therefore, more effort should be devoted to the comprehensive understanding of ruminants' gut microbiota to find effective strategies and improve animal performance. On the other hand, 'omics' methods make it simple for us to develop new insight into the microbial component of ruminants' gut microbiota [4]. The main purpose of this study was to investigate the effects of silage treated with two common lactic acid bacteria inoculants (one homofermentative and the other heterofermentative) on the bacterial community and metabolites of the digestive tract of sheep.

## 2. Materials and Methods

### 2.1. Silage Making

Whole crop corn (*Zea mays* L.) at the half-milk-line stage (38.9% DM) was harvested in Ulanqab, Inner Mongolia. Then, the harvested material was immediately chopped to 1–2 mm by a forage cutter (Toyohira Agricultural Machinery, Sapporo, Japan). Three treatments were applied: a control without inoculation and two inoculations using *Lentilactobacillus plantarum* MTD/1 (Vita Plus, Madison, MI, USA) or *Lentilactobacillus buchneri* 40788 (Vita Plus, Madison, MI, USA). The inoculants were mixed into distilled water and applied at a rate of $1 \times 10^6$ cfu/g fresh matter (FM), and an equal volume of distilled water was sprayed in the fresh corn as a control. The two inoculations, *Lentilactobacillus plantarum* MTD/1 (P) and *L. buchneri* 40788 (B) (Vita Plus, Madison, MI, USA), were used at a rate of

$1 \times 10^5$ cfu g/g fresh matter (FM) according to the procedure described by Xu et al. [13]. The chopped whole crop corn was randomly divided into 30 batches to result in ten bags per treatment, and then the materials were packed into plastic bags ($760 \times 950$ mm, 0.1 mm in thickness). Air was removed by using a vacuum cleaner before the top of the bag was tied with nylon strings. Double plastic bags were covered with cloth bags to avoid damage during movement and storage. In total, 30 kg $\times$ 10 bags for each treatment were prepared. Silage was stored for approximately 8 months at ambient temperature.

### 2.2. Silage Quality Analysis

Silage simples were randomly collected from five bags of each treatment for the analysis of fermentation characteristics and aerobic stability, according to Han et al. [14]. About 100 g of material was oven-dried at 60 °C to determine the DM content. Twenty grams of silage was soaked with 180 mL distilled water, and the squeezed water was filtered. The pH was measured immediately using a pH meter (sartorius PB-10, Göttingen, Germany). Organic acids were measured using high-performance liquid chromatography, as described by Xu et al. [13]. The prepared filtrate was acidified and filtered with a 0.22 μm filter. The determination was performed on a KC-811 column with SPD-M10AVP as the detector. The mobile phase was determined with 3 mmol/L perchloric acid, the injection volume was 5 μL, the column temperature was 50 °C, the flow rate was 1 mL/min, and the detection wavelength was 210 nm. Approximately 100 g of representative silage sample from each bag was placed loosely into a plastic bucket. The temperature of each bucket was recorded every 15 min, and the ambient temperature was recorded from the empty plastic bucket. Buckets were exposed to air in the laboratory ($26 \pm 1$ °C). Aerobic stability was calculated as the number of hours before the temperature of the silage rose 2.0 °C above its ambient temperature.

### 2.3. Animal Experiments and Sampling

The dried corn stalk (S) was set as a negative control to compare the silage effects on rumen fermentation type and microbial communities. The corn stalk and the three corn stalk silages were mixed with concentrates, including cracked maize, soybean meal, and wheat bran (Table 1). The chemical composition indexes of the experimental feed were calculated according to the data of a common feed ingredient list. Twenty-four local hybrid rams (a hybrid of Small Tail Han sheep and Mongolian sheep, with an average initial BW $27.8 \pm 3.05$ kg) that had never been fed with silage, were randomly divided into 4 groups. The feeding experiment lasted for 28 days. During the 21-day preliminary period, the animals were fully adapted to the single-stall feeding environment. Dry matter intake was measured over the next 7 days of the trial period. Sheep were given feeds twice, at 8:00 and 17:00, every day with unlimited access to fresh water. Animal care for this experiment complied with the practices outlined in the Guide for the Care and Use of Agricultural Animals in Research and Teaching (FASS, 2010).

The rectal feces were collected at the end of the experiment. PP gloves were worn, and 2–3 g of fecal samples was collected from the rectum in a sterile centrifuge tube and brought back to the laboratory in a liquid nitrogen tank. After mixing well, 200 mg was taken to DNA extraction for fecal microbiological analysis. Then, the sheep were slaughtered at Baimiaozi Hohhot, Inner Mongolia. The collection of sheep rumen samples was based on Li et al. [15]. The rumen content of the sheep was first homogenized by hand using disposable polyethylene gloves. To obtain the liquid and solid samples, the whole rumen contents were strained through four layers of sterilized gauze. Approximately 40 mL of strained liquid and the remaining pellets, representing the solid fraction, were collected in sterilized tubes. The pH of the rumen fluid was immediately measured using a portable pH meter (PB-10 Sartorius). To obtain the epithelial samples, the rumen walls were rinsed with sterile saline solution. Epithelial samples from an approximately $1 \times 1$ cm area of the rumen epithelium were obtained by scraping with a sterilized glass slide. All samples were brought back to the laboratory with liquid nitrogen and stored at $-80$ °C before use.

**Table 1.** Ingredients and nutrient composition of the diet.

| Ingredient Composition (DM%) | S | C | P | B |
|---|---|---|---|---|
| Corn stalk | 40.00 | 0.00 | 0.00 | 0.00 |
| Corn stalk silage | 0.00 | 40.00 | 40.00 | 40.00 |
| Ground corn | 33.00 | 30.73 | 30.73 | 30.73 |
| Rice bran | 2.27 | 11.50 | 11.50 | 11.50 |
| Soybean meal | 6.00 | 5.00 | 5.00 | 5.00 |
| Cottonseed meal | 11.20 | 8.59 | 8.59 | 8.59 |
| Mineral powder | 0.59 | 0.74 | 0.74 | 0.74 |
| Molasses | 6.00 | 2.50 | 2.50 | 2.50 |
| Salt | 0.71 | 0.71 | 0.71 | 0.71 |
| Premix | 0.23 | 0.23 | 0.23 | 0.23 |
| Metabolic energy (MJ/kg DM) | 10.80 | 11.80 | 11.80 | 11.80 |
| CP (%) | 15.00 | 15.10 | 15.10 | 15.10 |
| Ca (%) | 0.45 | 0.49 | 0.49 | 0.49 |
| P (%) | 0.41 | 0.48 | 0.48 | 0.48 |
| NDF (%) | 28.25 | 31.02 | 31.02 | 31.02 |
| ADF (%) | 18.00 | 16.77 | 16.77 | 16.77 |

DM, dry matter content; CP, crude protein; NDF, neutral detergent fiber; ADF, acid detergent fiber; S, corn stalk; C, corn stalk silage; P, corn stalk silage treated with *Lentilactobacillus plantarum*; B, corn stalk silage treated with *L. buchneri*.

### 2.4. DNA Extraction PCR Amplification and 16S rDNA Sequencing

DNA from different samples (three replications for each treatment of silage and six replications for each treatment of digestive tract samples) was extracted using the Omega Stool DNA Kit (D4015, Omega, Inc., Norwalk, CT, USA) according to the manufacturer's instructions. The reagent, which was designed to uncover DNA from trace amounts of a sample, has been shown to be effective for the preparation of the DNA of most bacteria. Nuclease-free water was used as blank. The total DNA was eluted in 50 µL of elution buffer and stored at $-80\ ^{\circ}$C until measurement using PCR by LC-Bio Technology Co., Ltd. (Hangzhou, China). The V3–V4 region of the 16S rRNA gene was amplified with the primers 341F (5′-CCTACGGGNGGCWGCAG-3′) and 805R (5′-GACTACHVGGGTATCTAATCC-3′). PCR amplification was performed in a total volume of 25 µL of reaction mixture containing 25 ng of template DNA, 12.5 µL PCR premix, 2.5 µL of each primer, and PCR-grade water to adjust the volume. The PCR conditions to amplify the prokaryotic 16S fragments consisted of an initial denaturation at 98 $^{\circ}$C for 30 s; 32 cycles of denaturation at 98 $^{\circ}$C for 10 s, annealing at 54 $^{\circ}$C for 30 s, and extension at 72 $^{\circ}$C for 45 s; and then final extension at 72 $^{\circ}$C for 10 min. The PCR products were purified and quantified, and then they were sequenced using the Nova SeqPE250 platform. The samples were sequenced on an Illumina Nova Seq platform provided by LC-Bio according to the manufacturer's recommendations. Paired-end reads were assigned to samples on the basis of their unique barcode and truncated by cutting off the barcode and the primer sequence. Paired-end reads were merged using FLASH. Quality filtering on the raw tags was performed under specific filtering conditions to obtain the high-quality clean tags according to fqtrim (V 0.94). Chimeric sequences were filtered using Vsearch software (v2.3.4). Sequences with $\geq$97% similarity were assigned to the same operational taxonomic units (OTUs) by Vsearch (v2.3.4). Representative sequences were chosen for each OTU, and taxonomic data were then assigned to each representative sequence using the RDP (Ribosomal Database Project) classifier. OTU abundance information was normalized using a standard sequence number corresponding to the sample with the fewest sequences. Alpha diversity was applied to analyze the complexity of species diversity for a sample through 5 indices, namely Chao1, Observed species, Goods coverage, Shannon, and Simpson, and all indices in our samples were calculated with QIIME (v1.8.0). Beta diversity analysis was used to evaluate the differences between samples in species complexity. Beta diversities were calculated by PCA using QIIME software. Blast was used for sequence alignment, and the OTU representative sequences were annotated with the RDP (ribosome database) and

NCBI-16S databases for each representative sequence. Other diagrams were implemented using the R package (v3.4.4). The sequencing data were submitted to the NCBI with the BioProject accession number PRJNA612148.

### 2.5. Rumen Metabolites Analysis

The collected rumen liquid samples were thawed on ice, and metabolites were extracted with 50% methanol buffer. After centrifugation at $4000 \times g$ for 20 min and filtration through a 0.22 μm microspore filter membrane, the supernatants were analyzed using a UPLC-ESI-MS/MS system according to Li et al. [16]. In addition, pooled QC samples were also prepared by combining 10 μL of each extraction mixture. All samples were acquired by the LC–MS system following the machine's directions. First, all chromatographic separations were performed using an ultra-performance liquid chromatography (UPLC) system (SCIEX, Macclesfield, UK). An ACQUITY UPLC T3 column (100 mm × 2.1 mm, 1.8 μm, Waters, Wilmslow, UK) was used for reversed-phase separation. The column oven was maintained at 35 °C. The flow rate was 0.4 mL/min, and the mobile phase consisted of solvent A (water, 0.1% formic acid) and solvent B (acetonitrile, 0.1% formic acid). The gradient elution conditions were set as follows: 0–0.5 min, 5% B; 0.5–7 min, 5% to 100% B; 7–8 min, 100% B; 8–8.1 min, 100% to 5% B; 8.1–10 min, 5% B. The injection volume for each sample was 4 μL. The high-resolution tandem mass spectrometer TripleTOF 5600 plus (SCIEX, UK) was used to detect metabolites eluted from the column. The Q-TOF was operated in both positive and negative ion modes. The curtain gas was set to 30 PSI, ion source gas 1 was set to 60 PSI, ion source gas 2 was set to 60 PSI, and the interface heater temperature was 650 °C. For positive ion mode, the IonSpray voltage floating was set at 5000 V. For negative ion mode, the IonSpray voltage floating was set at −4500 V. The mass spectrometry data were acquired in IDA mode. The TOF mass range was from 60 to 1200 Da. The survey scans were acquired in 150 ms, and as many as 12 product ion scans were collected if exceeding a threshold of 100 counts per second (counts/s) and with a 1+ charge-state. The total cycle time was fixed at 0.56 s. Four time bins were summed for each scan at a pulser frequency value of 11 kHz by monitoring the 40 GHz multichannel TDC detector with four-anode/channel detection. Dynamic exclusion was set to 4 s. During the acquisition, the mass accuracy was calibrated every 20 samples. Furthermore, in order to evaluate the stability of the LC–MS during the whole acquisition, a quality control sample (pool of all samples) was acquired after every 10 samples.

### 2.6. Statistical Analysis

The influence of inoculants on fermentation characteristics and microbial composition was analyzed by one-way analysis of variance (ANOVA) (IBM SPSS 20.0) using the following model:

$$Y_{ij} = \mu + T_i + e_{ij}$$

where $Y_{ij}$: dependent variable, $\mu$: overall mean, $T_i$: effect of treatment (I = 1–3), $e_{ij}$: error term. The differences among treatment means were assessed by Duncan's new multiple range test, and the significance level was accepted at $p < 0.05$. Principal component analysis, an unsupervised chemometric method, was used to assess the presence of any clustering, trends, or outliers. Canonical correlation analysis was used to analyze the correlation between main bacterial communities and the detected metabolites.

## 3. Results and Discussion

Under anaerobic conditions, soluble carbohydrates in raw materials are fermented to organic acids by lactic acid bacteria during ensiling. As a result, lactic acid bacteria become dominant, and other undesirable microorganisms are inhibited [17]. The dynamic changes in microbial species and numbers during ensiling are important indicators that affect the fermentation quality and feed nutrients of silage [18,19]. As shown in Table 2, the addition of P increased the DM content compared with the control and B-treated silage. Although a meta-analysis concluded that lactic acid bacteria inoculation of corn silages

had no effects on silage DM concentration ($p$ = 0.87) [8], Kung and Muck [20] reported that silages treated with inoculants such as P often exhibit better DM recovery compared with untreated silages, and our present results agree with their findings. Muck et al. [21] reported that homofermentative LAB often rapidly decreases pH and increases lactic acid relative to other fermentation products. During ensiling, lactic acid (pKa of 3.86) produced by LAB is usually the acid found in the highest concentration in silages, and it contributes the most to the decline in pH during fermentation because it is about 10 to 12 times stronger than any other major acid (e.g., acetic acid (pKa of 4.75) and propionic acid (pKa of 4.87)) found in silages [22]. However, a meta-analysis indicated no reduction in pH ($p$ = 0.39) in corn silages compared with untreated silages. It was also confirmed by our results that there was no significant difference in pH between the P- and B-treated groups and the control group. A heterofermentative lactic acid bacterium, B has been reported as an additive to improve the aerobic stability of silages [23] via the anaerobic conversion of moderate amounts of lactic to acetic acid. Our results are partially consistent with this; the contents of lactic acid and acetic acid in both the P- and B-treated groups were not significantly different from those of the control group. However, the addition of B tended to increase the aerobic stability compared with P-treated silage. The addition of P increased the lactic acid bacteria number of corn stalk silage, though not significantly. Similar results have been reported by Guo et al. [24] and Xu et al. [13]. As shown in Figure S1, *Bacillus* was the dominant genus in corn stalk, and *Lentilactobacillus* became the most abundant after ensiling. Xu et al. [13] also reported that *Lentilactobacillus* became the dominant bacterial community after 90 days of whole corn silage fermentation, which was consistent with our results. Lactic acid bacteria are important for energy and protein preservation during the ensiling process [21]. Lactic acid bacteria strains belonging to *Lentilactobacillus*, *Pediococcus*, *Enterococcus* are always applied during silage production to accelerate pH decline. A decline in pH and an increase in the abundance of acid-tolerant *Lentilactobacillus* inhibit undesirable microorganisms and thus reduce nutrient loss during ensiling [25]. As expected, the addition of *L. plantarum* and *L. buchneri* increased the relative abundance of *Lentilactobacillus* (Figure S1). This indicates that the addition of lactic acid bacteria inoculants has positive effects on corn stalk silage.

**Table 2.** Fermentation quality and aerobic stability of corn stalk silages.

| Item | C | P | B | SEM | *p*-Value |
|---|---|---|---|---|---|
| DM (%) | 32.1 b | 37.0 a | 32.9 b | 0.80 | 0.001 |
| pH | 3.89 | 3.93 | 3.86 | 0.01 | 0.089 |
| LAB (log cfu/g) | 3.60 | 5.28 | 4.11 | 0.35 | 0.128 |
| Coliform bacteria (log cfu/g) | <2.00 | <2.00 | <2.00 | <0.01 | 1.00 |
| Lactic acid (%DM) | 5.55 | 3.64 | 5.39 | 0.99 | 0.737 |
| Acetic acid (%DM) | 2.00 | 1.45 | 2.28 | 0.44 | 0.788 |
| Aerobic stability (h) | 68 | 36 | 128 | 17.9 | 0.081 |

DM, dry matter content; LAB, lactic acid bacteria; S, corn stalk; C, corn stalk silage; P, corn stalk silage treated with *Lentilactobacillus plantarum*; B, corn stalk silage treated with *L. buchneri*; SEM, standard error of means. Values with different letters (a,b) show significant differences among treatments.

To investigate the effects of the materials above on the digestive tract microbiota of sheep, the bacterial community of feces and different parts of the rumen were analyzed. As shown in Figure 1a, principal component analysis indicated that the bacterial community of different parts of the rumen was not differentiated. This suggests that each of the three parts could be sampled for rumen bacterial community analysis in the future. A small difference was found in the bacterial community between the rumen and feces. At the phylum level, Firmicutes and Bacteroidetes were two dominant phyla in the digestive tract microbiota of sheep (Figure 1b), which is consistent with the results reported by Zhang et al. [9]. Compared with the bacterial communities in feces, the abundance of Firmicutes was lower and that of Bacteroidetes was higher in the rumen. This is consistent with Mu et al. [26], who speculated that the extra Bacteroidetes present in rumen fluid may be enriched for cellulolytic bacteria, as the fiber content in the rumen was much higher than that

in the hindgut. They also found that the relative abundance of Proteobacteria was higher in high-production dairy cows than that in the low-yield group. In the present study, a higher abundance of Proteobacteria was observed in sheep fed with silages than in those fed with corn stalk. P-treated silage decreased the relative abundance of Firmicutes in the rumen fluid. Similar results were reported by Monteiro et al. [27], who fed high-producing dairy cows with *Lentilactobacillus plantarum* directly. The diet has a great influence on the rumen microbiota. Compared with corn stalk, sheep fed with corn stalk silage showed a higher relative abundance of *Prevotella* and a lower relative abundance of Ruminococcaceae_unclassified, Bacteroidetes_unclassified, and Firmicutes_unclassified in their rumen and feces (Figure 1c). The addition of *Lentilactobacillus plantarum* decreased the relative abundance of *Prevotella* and *Butyrivibrio* and increased that of *Bifidobacterium*, *Treponema*, *Saccharofermentans*, and *Ruminococcus* in the rumen (Figure 2). *Lentilactobacillus buchneri*-inoculated silage increased the relative abundance of *Prevotella* and *Ruminococcus*. It is known that *Prevotella*, *Ruminococcus*, and *Fibrobacter* in the digestive tract are involved in fiber metabolism. The relative abundance of these genera increased in the lactic acid bacteria-treated silage groups in the present study. Similarly, Chen et al. [28] and Chen et al. [29] found that alfalfa silage inoculated with *L. plantarum* showed higher quantification of *Prevotella* and activity of cellobiase, carboxymethyl-cellulase, pectinase, and protease than the untreated silage and thus enhanced the digestibility of DM, crude protein, and neutral detergent fiber using an in vitro method. *Bifidobacterium* is a common probiotic that can compete with pathogenic bacteria. The higher *Bifidobacterium* in the P group than in the control group indicated that the P-inoculated silage might also have a positive effect on ruminants. However, it is difficult to explain the lower *Bifidobacterium* abundance in untreated corn stalk silage compared with that in the corn stalk group. It might be because of the lower pH value in the corn stalk silage group (Table 2), as the biological activity and growth declined with the drop in pH value [30].

The pH value and metabolites of ruminal fluid are essential for the health and metabolism of the host and can be influenced by the rumen microflora. The results of pH and volatile fatty acid content analysis are shown in Table 3, and the metabolites analysis is shown in Figure 3 and Tables S1–S3. It is known that the rumen is a complex microecosystem with large numbers of microorganisms, and the environment is relatively stable. In the present study, significant differences between the control and the treatments were not observed. This might be because the slight change in corn silage induced by lactic acid bacteria inoculants was insufficient to change rumen fermentation.

**Table 3.** pH and volatile fatty acid content in the rumen fluid.

| Item | S | C | P | B | SEM | *p*-Value |
|---|---|---|---|---|---|---|
| Acetic acid (mg/mL) | 2.46 | 2.63 | 2.32 | 2.19 | 0.15 | 0.785 |
| Propionic acid (mg/mL) | 0.66 | 0.66 | 0.62 | 0.65 | 0.05 | 0.989 |
| Butyric acid (mg/mL) | 0.34 | 0.44 | 0.39 | 0.36 | 0.04 | 0.803 |
| pH | 7.26 | 6.73 | 6.94 | 6.83 | 0.09 | 0.157 |

S, rumen fluid of sheep fed corn stalk; C, rumen fluid of sheep fed corn stalk silage; P, rumen fluid of sheep fed corn stalk silage treated with *Lentilactobacillus plantarum*; B, rumen fluid of sheep fed corn stalk silage treated with *L. buchneri*; SEM, standard error of means.

A total of 8287 metabolites were detected in the rumen liquid, including lipids and lipid-like molecules, organic acids, and derivatives, which is consistent with the study by Wang et al. [26]. The principal component analysis of the metabolites showed that the variation among the biological replicates was small, and the six replicates of each group were clustered. This indicates the sufficient reliability and reproducibility of the experiment. Comparing corn stalk with corn stalk silage and silage treated with P or B revealed a significant difference. This indicates that silage and lactic acid bacteria inoculants during ensiling have a great influence on ruminal metabolites. The volcano plot analysis showed that a large number of differential metabolites was detected when comparing different groups. For example, a total of 498 differential metabolites (266 chemicals were

downregulated and 232 were upregulated) were detected when comparing corn stalk and corn stalk silage. This might be because of the activities of microorganisms and enzymes during fermentation, which result in chemical changes before and after ensiling. Previous studies have also reported that different chemical composition in diets results in different fermentation in the rumen [31,32]. A total of 257 differential metabolites were detected when comparing the untreated silage with silages treated with P, and 141 were detected when comparing the untreated silage with silage treated with B. This indicates that lactic acid bacteria inoculants used in silage production have an influence on rumen fermentation. Similar results were reported by Hu et al. [33].

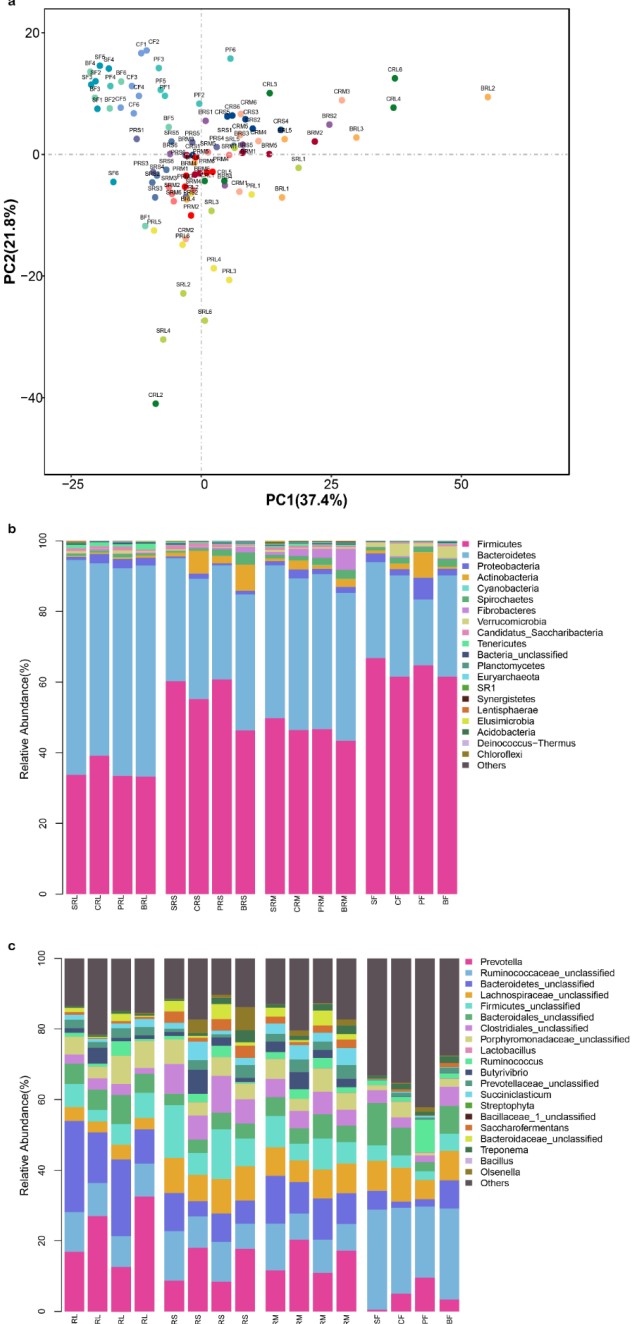

**Figure 1.** Principal component analysis (**a**) and the relative abundance at phylum (**b**) and genus (**c**) levels of bacterial communities in different parts of rumen (RL, rumen liquid; RL, rumen state; RM, rumen mucosa) and feces (F) of sheep fed with corn stalk (S), corn stalk silage (C), and corn stalk silage treated with *Lentilactobacillus plantarum* (P) or *L. buchneri* (B).

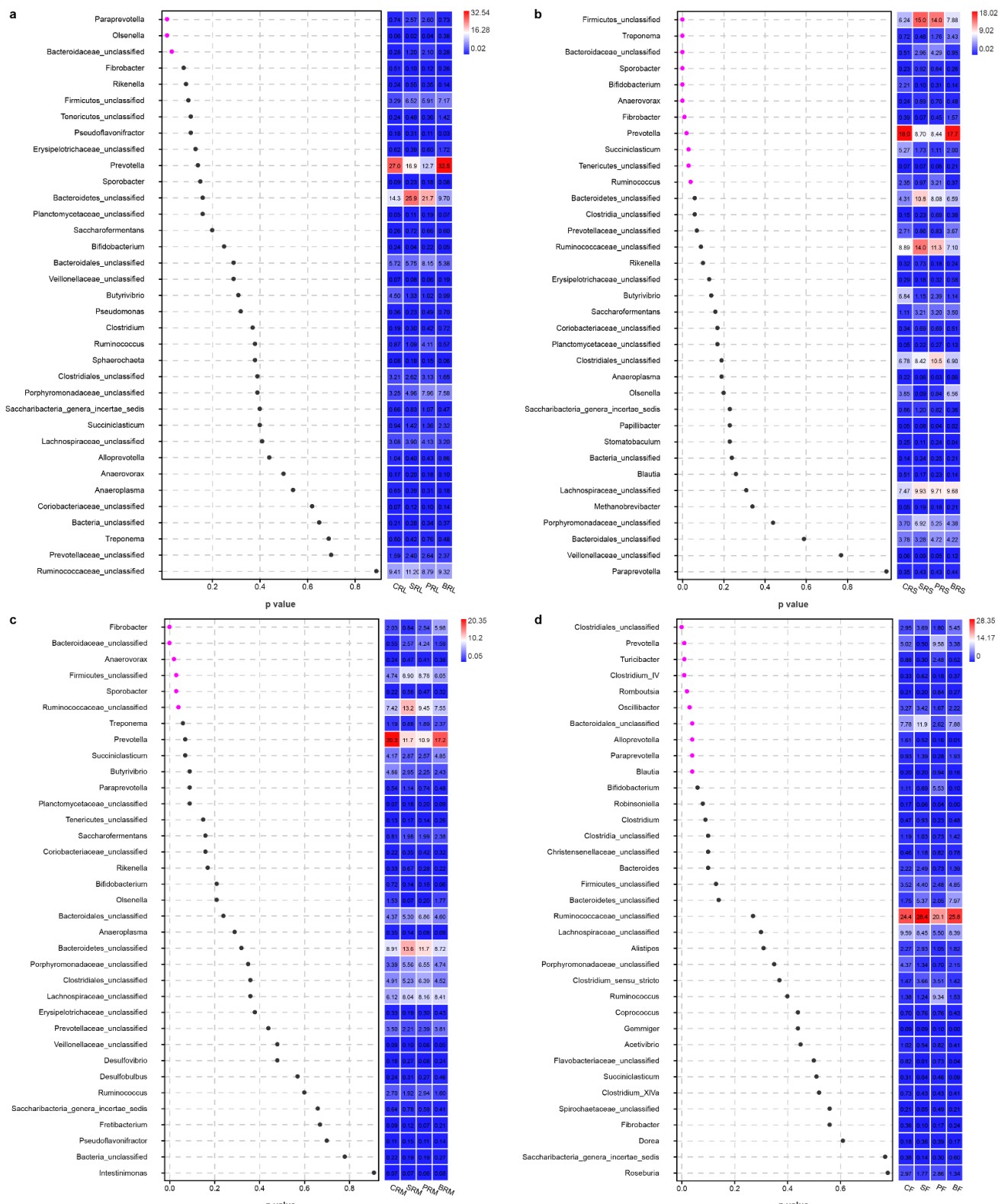

**Figure 2.** Difference analysis of bacterial communities in different parts of rumen ((**a**), RL, rumen liquid; (**b**), RL, rumen state; (**c**), RM, rumen mucosa) and feces ((**d**), F) of sheep fed with corn stalk (S), corn stalk silage (C), and corn stalk silage treated with *Lentilactobacillus plantarum* (P) or *L. buchneri* (B).

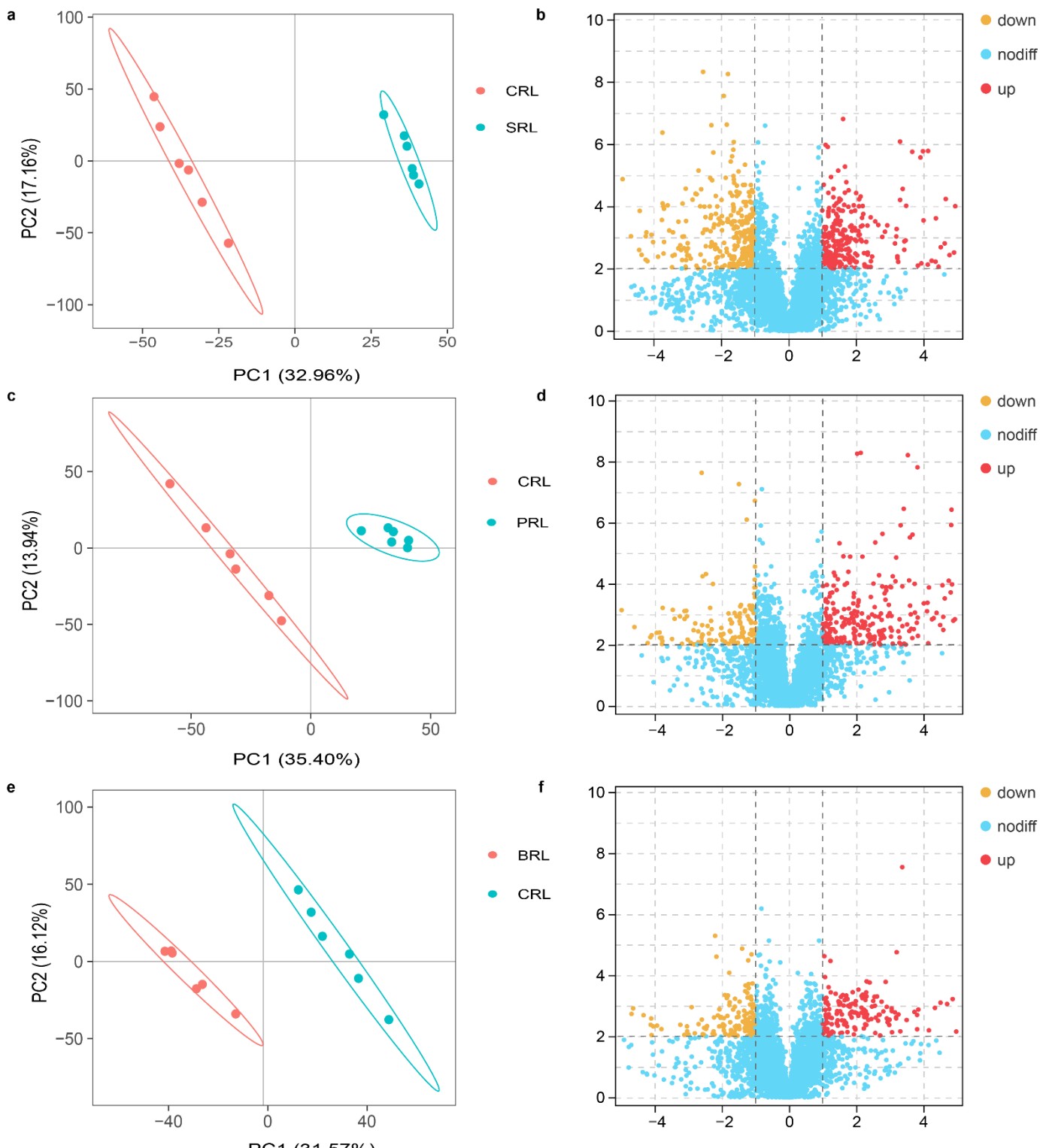

**Figure 3.** Difference analysis of metabolites in rumen liquid (RL) of sheep fed with corn stalk (S), corn stalk silage (C), and corn stalk silage treated with *Lentilactobacillus plantarum* (P) or *L. buchneri* (B) ((**a**,**c**,**e**), principal component analysis of S vs. C, C vs. P, and C vs. B, respectively; (**b**,**d**,**f**), volcano plot analysis of S vs. C, C vs. P, and C vs. B, respectively). PCA1 described 32.96%, 35.40%, and 31.57% of the variance, and PCA2 described 17.16%, 13.94%, and 16.12% of the variance in the negative ion mode of a, c, and e, respectively.

Rumen digestion is known as a fermentation process based on microorganisms. Canonical correlation analysis of bacterial and metabolites suggested that these metabolites in the sheep rumen were correlated with bacterial communities, especially *Butyrivibrio*, *Fibrobacter*, and *Prevotella* (Figure 4). Similarly, Li et al. [16] found that some metabolites in the ruminal fluid were associated with different levels of microbiota. The correlation between the bacterial community and metabolites indicates that altering the bacterial community might be an effective way to promote rumen digestion. *Butyrivibrio* spp. is considered a genus with the ability to degrade cellulose completely and quickly and produce a variety of monosaccharides and acids [34]. Lima et al. [35] found that the abundance of *Butyrivibrio* was positively correlated with milk yield. This may partially explain the correlation between the relative abundance of *Butyrivibrio* and metabolites in the rumen fluid. *Prevotella* and *Fibrobacter* are known as plant cell wall, starch, and protein utilization bacteria, and they can produce short-chain fatty acids, such as acetate, propionate, and propionate-precursor succinate. They play important roles in the conversion of nutrients in the rumen, and this may partially explain the correlation between their relative abundances and metabolites in the rumen fluid. Similar results were reported by Wang et al. [25]. Similarly, the members of some genera, such as *Saccharofermentans*, *Paraprevotella*, *Succiniclasticum*, *Ruminococcus*, and *Saccharibacteria,* are also important in feed digestion in the rumen, which might lead to the positive correlation between their relative abundances and metabolites in the rumen fluid.

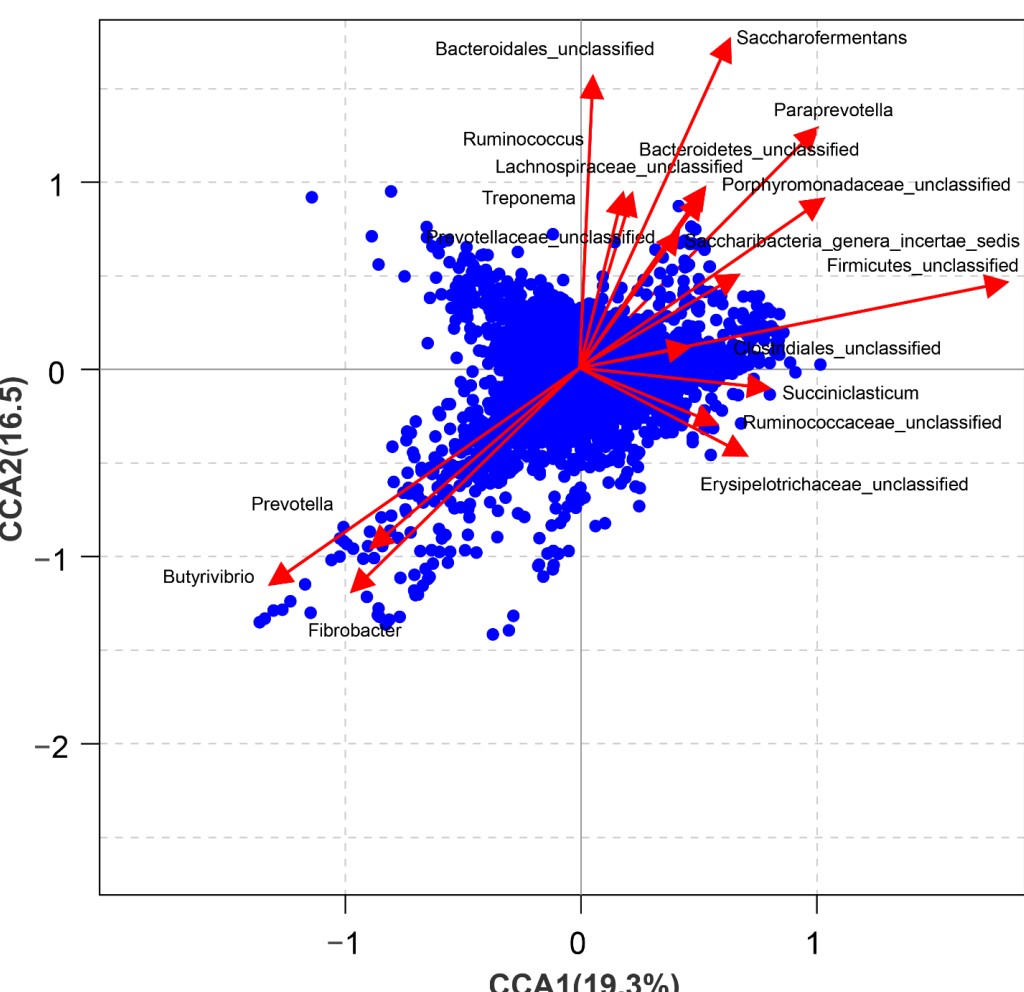

**Figure 4.** Canonical correlation analysis of bacterial communities and metabolites in the rumen liquid.

## 4. Conclusions

In summary, the addition of *Lentilactobacillus plantarum* MTD/1 and *L. buchneri* 40788 during ensiling not only showed positive effects on silage quality by altering the bacterial community but also changed the bacterial community and metabolites in the digestive tract of ruminants. The metabolites in the rumen were correlated with the bacterial communities. The lactic acid bacteria inoculants in silage production might be beneficial for animal performance and health by influencing the bacterial community and metabolites in the digestive tract of ruminants.

**Supplementary Materials:** The following supporting information can be downloaded at: https://www.mdpi.com/article/10.3390/fermentation8070320/s1. Figure S1. Bacterial communities of corn stalk (S), corn stalk silage (C), and corn stalk silage treated with *Lactobacillus plantarum* (P) and *L. buchneri* (B). Table S1. Differential metabolites of ruminal liquid between the corn stalk (S) and corn stalk silage (C). Table S2. Differential metabolites of ruminal liquid between the corn stalk silage (C) and corn stalk silage treated with *Lactobacillus plantarum* (P). Table S3. Differential metabolites of ruminal liquid between the corn stalk silage (C) and corn stalk silage treated with *L. buchneri* (B).

**Author Contributions:** Writing and data curation, H.H.; investigation and data curation, H.H. and C.W.; supervision, H.H., Y.X. and X.G.; writing—review and editing, H.H., C.W. and Y.X.; project administration, funding acquisition, and writing—review and editing, H.H.; formal analysis and methodology, Z.H., L.S. and Y.Z. All authors have read and agreed to the published version of the manuscript.

**Funding:** This study was funded by the National Natural Science Foundation of China, NSFC (project no. 31902192), and the Science and Technology Major Project of the Inner Mongolia Autonomous Region of China (project no. zdzx2018065).

**Institutional Review Board Statement:** This study followed the recommendations of the Inner Mongolia University Animal Care and Use Committee (Record No. IMU-SHEEP-2020-035).

**Informed Consent Statement:** Not applicable.

**Data Availability Statement:** All 16S rRNA sequences were submitted to the National Center for Biotechnology Information (NCBI), Temporary Submission ID: SUB7118893.

**Conflicts of Interest:** The authors declare that they have no conflict of interest.

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
