# Peer review of "Effects of Lactic Acid Bacteria-Inoculated Corn Silage on Bacterial Communities and Metabolites of Digestive Tract of Sheep"

_fermentation, doi:10.3390/fermentation8070320_

Round 1

Reviewer 1 Report

Manuscript ID: Fermentation 1759917

Effects of lactic acid bacteria inoculated silage on bacterial communities and metabolites of digestive tract of sheep

by Han et al.

General comments

The objective of the research presented in this manuscript was to investigate the effect of two different silage bacterial additives on the rumen microflora and metabolites of sheep.

This project would provide additional information on the impact of silage microflora on rumen microbiology. Although few studies directly testing the impact of silage microflora, and the importance of its population of lactic acid bacteria are few, several articles were published on the impact of different feeds on rumen microbial diversity and activities.

The authors should consult the existing literature in relation pertaining to the subject to expand the introduction and the discussion. They should include this journal article on the importance of lactic acid bacteria in the rumen – Yang et al. DOI: 10.3389/fmicb.2018.00718.

The authors should:

-       Expand introduction to explain and justify the choice of the lactic acid bacteria used

-       Improve the methodology on (1) clearly define what are the treatment used, (2) analysis of the sequencing results – the bioinformatic part, (3) explain why they chopped the forage to tiny pieces between 1-2 mm, (4) expand on the metabolites analysis, (5) expand the statistical analysis section to include description of the model for ANOVA and all the Principal Component Analysis performed, (6) provide more details on collecting the different rumen samples, (7) provide details on collecting the feces before killing the animals

-       Expand the discussion in relation to the microbial diversity of the silage, microbial diversity of the rumen, and metabolites in the rumen.

-       Provide high quality figures. All four panels of Figure 2 are unreadable.

-       Explain why the PCA for Figure 3 were performed separately in not by combining all treatments together.

Specific comments

­­

The authors should be aware that the taxonomy of the genus Lactobacillus was amended in 2020. See Zheng et al. DOI : 10.1099/ijsem.0.004107

Reviewer 2 Report

ID: fermentation-1759917

Title: Effects of lactic acid bacteria inoculated silage on bacterial communities and metabolites of digestive tract of sheep

The goal of this research was to see how two typical lactic acid bacteria inoculants (L. plantarum and L. buchneri) treated silage affected the bacterial community and metabolites in sheep's digestive tract. Although the work is intriguing, there are several points that need to be addressed.

Title: “...silage...” >>> corn silage / It should be clearly specified as corn silage because this study used corn silage as the main roughage.

Abstract:

-L17. “...digestive tract, 24 local hybrid...” >>> digestive tract.  Twenty-four local hybrid...

- “Lactobacillus plantarum (LP) and L. buchneri (LB)” The abbreviations used are not the same as shown in the tables and figures / Please check / The authors should edit them to be the same for all words.

- Lack of mention of the effect of adding LAB on the silage quality.

Introduction:

-L48: "...its good palatability, high digestibility of livestock, and low cost” Really? But I don't think that's the main purpose of making silage. Its main purpose should be to prolong the shelf life and maintain the quality of feed in order to be able to raise animals in shortage times.

/ It is highly digestible. When you compare with what? / Is it really low cost? If you have to add more steps or processes to make the silage and may need more labor.

-L49: “(McDonald et al. 1991)...” Please check the in-text reference style.

-L57: “Zhang et al. (2019)...” >> > Zhang et al. [no.] /Please check the in-text reference style.

-L59: “Cherdthong et al. (2021)...”  Not found in the reference list. / Please check the in-text reference style.

- Please check all in-text reference styles

- Why did the authors choose to study the effect of L. plantarum and L. buchneri?

Materials and Methods:

-L84: “Silos were stored for approximately 8 months...” Do the authors silage in plastic bags or silos? / How long does it take for the fermentation process? / Is it taking too long? /Why does it take 8 months? /Basically, the fermentation process takes 10 days to 3 weeks to complete for silage.

- Lack of feed chemical composition analysis procedures (CP, P, Ca, NDF, ADF, metabolic energy). Please specify those analysis procedures also.

- Lack of the ruminal VFAs analysis procedures (C2, C3, C4). Please specify those analysis procedures also.

-L91: “...high-performance liquid chromatography” Please specify the model (model, branding, manufacture country) of the HPLC, the column, and the mobile phase used in this study.

-L96: “24 local hybrid sheep…” >>> Twenty-four local hybrid sheep...

-L98: “...a 21-day preliminary period and a 7-day trial period.” Do the authors mean this study was 28 days experiment? / Do the authors mean the first 21 days of animal adaptation and the last 7 days for sample collection? / If yes, please add more details.

-L106: How many sheep were slaughtered? / Please specify the slaughterhouse location.

-L137: “UPLC-ESI-MS/MS system according to Hu et al. (2020)” In the report of Hu et al. (2020), they used the GC–TOF/MS system for rumen metabolites analysis in their study, it was not UPLC-ESI-MS/MS. / Please check.

Results & Discussion:

-L144-146: Those sentences are unclear, and they should be revised as well as citations added. Recommend that the authors change and add citations as needed. “Under anaerobic conditions, soluble carbohydrates in raw materials are fermented to organic acids by lactic acid bacteria during ensiling. As a result, lactic acid bacteria take dominance and other undesirable microorganisms are inhibited (So et al., 2020; Kaewpila et al., 2021)” See detail at https://doi.org/10.1080/10495398.2020.1781146  and https://doi.org/10.1038/s41598-021-81505-z

-L153: “(Figure S1)” Is it Figure S1 or Figure 1?

-L155, 156: What are LB and LP? / Please specify abbreviations and meanings in the materials and methods section.

- L155,156: Why did the addition of LB and LP not significantly increase the number of lactic acid bacteria? / Please explain more discussion about the reasons.

-L198, 199: Please check the P-value of pH! It didn't show significant differences (P=0.08).

- From the table2, the LAB looks like an increase with P and B. but why did it not drop in the pH or increased the lactic acid concentration? / Please explain more discussion about the reasons.  

- Lack of description of DM in silage results. / What's happened in the DM of silage that significant difference? / Please explain more discussion about the reasons.

- Lack of description of ruminal pH and VFAs content. / What's happened in the ruminal pH and VFAs content? / Please explain more discussion about the reasons as to why those parameters were similar.      

-L233: “257 and 141 differential... respectively” Don’t understand.

-L234: “...LP, LB...” >>> LP and LB

-L235: “...have an influence on rumen fermentation” How did lactic acid bacteria inoculants have an influence on rumen fermentation? / Which type has a more positive or negative effect on rumen fermentation metabolite?

-L254: “...Ruminococcus, Saccharibacteria...” >>> Ruminococcus, and Saccharibacteria

Conclusions:

-L261: “...improved silage quality...”?  The results of table2 were shown in similar results (no significant difference impact)

-L261-262: “...changed the bacterial community and metabolites in the digestive tract of ruminants” When comparing the results of L. plantarum and L. buchneri., Are both inoculants show different results or not?  How did both inoculants affect silage quality improvements, changes in bacterial communities, and metabolites in the digestive tract? (positive or negative)

-As this study results, Is it still necessary to add LAB inoculants to the silage? Because Table 2 of the results (silage quality) were shown no different impact between the addition of LAB inoculants group and basic corn silage. In addition, the ruminal end-products (VFAs) (Table 3) also show similar results between the addition of LAB inoculants group and basic corn silage.

References:

-Please check all in-text reference styles

- Not found some citations in the reference list

- Please check the journal reference style

Table2:

-Please check the P-value of pH and aerobic stability value (P=0.08) / Why did the author show the superscript in those parameters? /Are there significant differences?

-Please add the meaning of superscript (a, b, c) in the footnote.

- From the table2, the LAB and lactic acid data look like significant differences/ Why did those data show a lot of the SEM? / Does it mean this parameter study analysis was too much error or not? / Please check raw data.

-L160: It has no S, corn stalk in table2.

Table3:

- Please check the P-value of ruminal pH (P=0.15) / Why did the author show the superscript in those parameters? /Are there significant differences?

-Why did table 3 have shows the parameters of S, but it has not in Table 2?

Figure 1a, 3:

-What are PC1 and PC2? / Please add the word meaning in the figure description

Figure 4:

- What are CCA1 and CCA2? / Please add the word meaning in the figure description

Round 2

Reviewer 2 Report

Fermentation-1759917

Effects of lactic acid bacteria inoculated silage on bacterial communities and metabolites of digestive tract of sheep

Han Et al.

Even though the authors revised the previous comments accordingly, some of them were required to be modified.

Title

-Each word must start with a capital letter.

Abstract

-L17: provide details of animal use, e.g., sex, initial BW, and breed.

-L30: Specific species of LAB that provide the potential recommendation.

Introduction

-L76-80: Add relevant citation.

Materials and methods

-L87-89: How old of plants?

-L118: Provide a methodology to assess aerobic stability.

-Table 1: Change salt to Salt

-L161-164: How many replications for DNA extraction for each sample? describes

-L198: Provide methodology on how rumen fluid was sampled. Rumen fistula or suction?

Results and discussion

-L249: Check citation style

-L252: Provide mode of action: why does LAB often rapidly decrease pH?

-L271-274: Provide a biological mechanism to support this state.

-L305: What do you mean "The diet has a great influence on the rumen microbiota" Please discuss!

-L366: Please clarify what metabolites of ruminal fluid that con be affected rumen microflora?

-L354-356: The sentences are unclear. What are 8000 metabolites? Please specify or revise the sentence.

L359-360: Suggest a change to "Comparing corn stalk with corn stalk silage, as well as silage that had been P or B treated, revealed a significant difference."

-L360-362: Provide a biological mechanism to support this state.

L391: What kind of digestion? Fiber? starch? Please make it clear!

-L393-394: Check error!

Conclusion

Please conclusion relating to the objective research. The strain of LAB should be indicated.

References

Check format following journal guidelines.
